# Preventive practices toward sexually transmitted infections and their determinants among young people in Ethiopia: A protocol for systematic review and meta-analysis

Etsay Woldu Anbesu[1]*, Setognal Birara Aychiluhm[1], Mussie Alemayehu[2]

**1** Department of Public Health, College of Medicine and Health Sciences, Samara University, Samara, Ethiopia, **2** School of Public health, College of Health Science, Mekelle University, Mek'ele, Tigray, Ethiopia

* etsaywold@gmail.com

**Funding:** The author(s) received no specific funding for this work.

## Abstract

### Background

Globally, the estimated annual number of new cases of curable sexually transmitted infections occurring among young people aged 15–24 years is approximately 178.5 million. There are fragmented and inconsistent findings on preventive practices for sexually transmitted infections. Thus, this systematic review and meta-analysis protocol aimed to estimate the pooled prevalence of preventive practices of sexually transmitted infections and identify its determinants among young people in Ethiopia.

### Methods

The Preferred Reporting Items for Systematic Review and Meta-analyses (PRISMA) will be used to develop the review protocol. Online databases such as PubMed, CINAHL, Scopus, Google, and Google Scholar will be used to search published and unpublished studies. The Joanna Briggs Institute Meta-Analysis of Statistics Assessment and Review Instrument will be used to assess the quality of the study. Statistical heterogeneity will be checked using the Cochran Q test and $I^2$ statistics. Subgroup analysis and meta-regression will be performed to identify the sources of heterogeneity. The statistical analysis will be performed using STATA version 14 software. A random-effects model will be performed to estimate the pooled prevalence and identify determinants of preventive practices of sexually transmitted infections.

### Discussion

Young people have a high unmet need for sexual and reproductive health services and poor preventive practices toward sexually transmitted infections. Although there are studies on preventive practices for sexually transmitted infections, there is no study finding on the pooled prevalence of preventive practices for sexually transmitted infections and its determinants among young people in Ethiopia. Thus, this systematic review and meta-analysis protocol will help to develop appropriate strategies.

**Competing interests:** The authors have declared that no competing interests exist.

## Introduction

Sexually transmitted infections (STIs) are diseases such as gonorrhea, syphilis, chancroids, lymphogranuloma venerum, and more than 30 different bacteria, viruses, and parasites. STIs can be curable and incurable. The curable STIs include gonorrhea, syphilis, trichomonas, and chlamydia. The incurable STIs include herpes simplex virus, hepatitis B, human immunodeficiency virus (HIV), and human papilloma virus (HPV). STIs are transmitted through sexual contact, such as vaginal, anal, and oral sex. It can also spread through nonsexual means via blood or blood products and mother to fetus during pregnancy, childbirth, and breastfeeding. The common symptoms or syndromes of STIs include urethral discharge, vaginal discharge, genital ulcer, lower abdominal pain, inguinal bubo, neonatal conjunctivitis, and scrotal swelling [1–4].

Globally, the estimated annual number of new cases of curable STIs that occur among people aged 15–49 years is 357 million, and approximately half of them are between 15–24 years [5, 6]. Every day, more than 1 million STIs are acquired [1]. STIs are health threats to adolescents and young people in developed and developing countries [4, 7–11]. In Ethiopia, although there is a lack of surveillance data, the self-reported prevalence of STIs among female and male youth aged 15–24 years was 3% and 1%, respectively [12]. STIs cause serious consequences, including an increased risk of HIV infection, stillbirth, neonatal death, low birth weight, sepsis, pneumonia, and neonatal conjunctivitis or blindness [1].

The age of young people by itself is a risk factor for many factors. It is a critical developmental period where youth begin to know and explain their sexual values and behaviors. They are at high risk for unsafe sexual behaviors, including STIs, unplanned pregnancy, abortion, low school performance, psychosocial problems, and economic crises [1, 13]. Moreover, rapid reproductive maturity among young people could lead to early sexual initiation and unsafe sex with the reluctance to use contraceptive methods [14–17].

In addition, factors such as multiple sexual partners, engaging in risky sexual behavior (RSB), sex without condoms, sex with older partners, consumption of alcohol and illicit drugs, cultural, religious, peer pressure, watching pornography, being single, nondisclosure of HIV status, and conflicts between couples and families affect young people's sexual behavior [4, 8, 10, 18–24].

In Ethiopia, although STIs remain one of the sustainable development goal (SDG) agendas [25] and the development of national guidelines for the management of STIs using a syndromic approach [26], there is a lack of attention and surveillance data on preventive practice toward STIs in young people [27]. Moreover, although studies have been conducted in different parts of the country on the preventive practices of STIs among young people, there are inconsistent findings on prevalence and its determinants. Thus, this systematic review and meta-analysis protocol aimed to estimate the pooled prevalence of preventive practices of STIs and identify its determinants among young people in Ethiopia.

### Research question

- What is the pooled prevalence of preventive practices toward STIs among young people in Ethiopia?

- What are the determinants of preventive practices toward STIs among young people in Ethiopia?

### Objectives

- To determine the pooled prevalence of preventive practices toward STIs among young people in Ethiopia

- To identify the determinants of preventive practices toward STIs among young people in Ethiopia

## Methods

### Review protocol development

The Preferred Reporting Items for Systematic Review and Meta-analyses (PRISMA) will be used to develop the review protocol [28], and the PRISMA-P 2015 checklist will be used to report the protocol procedures [29] (S1 File).

### PECO search guide

**Population.**  Young people (10–24 years old) [30].

**Exposure.**  Exposure is a determinant that increases or decreases the likelihood of preventive practices toward sexually transmitted infections.

**Comparison.**  The reference group for each determinant in each study will be the comparison variable. It may include good knowledge versus poor, positive attitude versus negative, education versus no education, access to information versus no accesses, consistent use of condoms versus not.

**Outcome.**  The outcome variable will be the pooled prevalence of preventive practices of STIs. Studies with the primary objective of determining the prevalence of preventive practices of STIs and their determinants among young people in Ethiopia will be considered.

### Data source and search strategies

Online databases such as PubMed, CINAHL, Scopus, Google, and Google Scholar will be used to search published and unpublished studies. As searching Google and Google Scholar leads to numerous studies, a limited number of studies will be screened from Google and Google Scholar using the phrase "Preventive practices toward Sexually Transmitted Infection in Ethiopia". The two authors (EW and SB) will retrieve the studies. In addition, across-reference search will be performed to add other related studies from the final included studies that may be missed during the database search. The search terms are indicated in S2 File. The search string will be adapted based on the specific requirements of the database to identify relevant studies. Retrieve studies will be exported to Endnote version 8 reference Manager software [31].

### Eligibility criteria

All observational studies (cross-sectional, case–control, and cohort) will be included in the systematic review and meta-analysis. Studies that reported the prevalence of preventive practices of STIs and its determinants among young people in Ethiopia will be included. Moreover, studies that reported only the prevalence of preventive practices of STIs or at least measured associations between determinant variables and the preventive practice of STIs will be considered. Institutional and community-based studies will be included. We will exclude studies that only address the qualitative approach. For both quantitative and qualitative data, only the quantitative data will be considered. We will not make restrictions on the date of publication. Studies published other than those in the English language, expert opinions, conferences, and case reports will be excluded.

## Selection of studies

The two authors (EW and SB) will independently screen the studies based on the titles and abstracts. Duplicate, irrelevant title, and abstract studies will be removed from the citation manager. The quality of full text studies will be evaluated, and studies that are not eligible will be removed. We will discuss with the third author (MA) to resolve any disagreement among reviewers during the review process. The selection process flow diagram will be presented using the PRISMA chart (S3 File).

## Quality assessment

To assess the quality and validity of the studies, the Joanna Briggs Institute Meta-Analysis of Statistics Assessment and Review Instrument (JBI-MAStARI) will be used [32]. The quality assessment will focus on clear inclusion criteria, study subjects and setting, standard measurement criteria, exposure and outcome measurements, and appropriate statistical analysis (S4 File). The quality of the studies will be assessed independently by the two authors (EW and SB). Studies 50% and above of the quality scale will be considered for the final systematic review and meta-analysis. During the quality review of the studies, any disagreement among reviewers will be resolved with the third author (MA).

## Data extraction

A data extraction template form on Microsoft Excel (2016) will be used. Before the beginning of the actual data extraction, we will pilot the Microsoft Excel data extraction form. The first author's name, publication year, study area, study design, sample size, associated factors, odds ratio, and prevalence of the studies will be included in the data extraction template. In addition, we will calculate the logarithm and standard error (SE) of the prevalence and odds ratio. The two authors (EW and SB) will extract the data independently. Discussion will be made for any difference with a third author (MA). We will contact the corresponding author of the studies in case of missing data or incomplete reports.

## Data synthesis and statistical analysis

Narrative synthesis of data will be done for the included studies. Summary tables and graphs will be performed to describe the characteristics of the included studies. Random-effects model [33] will be performed to estimate the overall pooled prevalence of preventive practices ofSTIs and identify its determinants among youths in Ethiopia. A 95% CI will be used to declare the statistical significance. Statistical heterogeneity will be checked using the Cochran Q test [34] and $I^2$ statistics [35]. $I^2$ values represent 25% low, 50% moderate, and 75% substantial heterogeneity. Subgroup nalysis and meta-regression will be done toidentify the sources of heterogeneity. Sensitivity analysis will be performed to assess the effect ofstudies on the overall estimation. The presence of publication bias will be checked usingfunnel plot [36], Egger's, and Beggar's test [37].

## Discussion

This systematic review and meta-analysis protocol aimed to estimate the pooled prevalence of preventive practices of STIs and identify its determinants among young people in Ethiopia. Although young people are sexually active, they have an unmet need for sexual and reproductive health services. The common barriers in low- and middle-income countries include lack of behavioral change and accessibility of services [38–40]. Despite different interventions

implemented to enhance the preventive practice of STIs among young people, the problem is still challenging in low-income countries, including Ethiopia [12, 26].

To the best of our knowledge, there is no study finding on the pooled prevalence of preventive practices of STIs and its determinants among young people in Ethiopia. Thus, this systematic review and meta-analysis protocol will help policymakers develop appropriate interventions on preventive practices of STIs in Ethiopia. This study protocol may have limited heterogeneity between studies. Only observational study designs published in the English language will be included.

## Supporting information

**S1 File. PRISMA-P 2015 checklist.**
(DOCX)

**S2 File. Draft of search strategy to be used using PubMed electronic database.**
(DOCX)

**S3 File. Diagrammatic presentation of the studies selection process for systematic review.**
(DOCX)

**S4 File. JBI critical appraisals for observational studies as shown in the link below https://jbi.global/critical-appraisal-tools.**
(DOCX)

## Acknowledgments

We would like to thank Samara University for the provision of free HINARY database websites, and internet and library access.

## Author Contributions

**Conceptualization:** Etsay Woldu Anbesu.

**Data curation:** Etsay Woldu Anbesu.

**Investigation:** Etsay Woldu Anbesu.

**Methodology:** Etsay Woldu Anbesu.

**Project administration:** Etsay Woldu Anbesu.

**Resources:** Etsay Woldu Anbesu.

**Supervision:** Mussie Alemayehu.

**Validation:** Setognal Birara Aychiluhm, Mussie Alemayehu.

**Visualization:** Etsay Woldu Anbesu, Setognal Birara Aychiluhm.

**Writing – original draft:** Etsay Woldu Anbesu.

**Writing – review & editing:** Etsay Woldu Anbesu, Setognal Birara Aychiluhm, Mussie Alemayehu.

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
