## [Decision Letter · Decision Letter 0]

19 Oct 2021

PONE-D-21-25527Preventive practices towards Sexually Transmitted Infection and its determinants among young people in Ethiopia: a protocol for a systematic review and Meta-analysisPLOS ONE

Dear Dr. Anbesu,

Thank you for submitting your manuscript to PLOS ONE. After careful consideration, we feel that it has merit but does not fully meet PLOS ONE’s publication criteria as it currently stands. Therefore, we invite you to submit a revised version of the manuscript that addresses the points raised during the review process.

We look forward to receiving your revised manuscript.

Kind regards,

Ali Rostami

Academic Editor

PLOS ONE

Journal Requirements:

3. We note that this manuscript is a systematic review or meta-analysis; our author guidelines therefore require that you use PRISMA guidance to help improve reporting quality of this type of study. Please upload copies of the completed PRISMA checklist as Supporting Information with a file name “PRISMA checklist

Reviewers' comments:

Reviewer's Responses to Questions

**Comments to the Author**

1. Does the manuscript provide a valid rationale for the proposed study, with clearly identified and justified research questions?

Reviewer #1: Yes

2. Is the protocol technically sound and planned in a manner that will lead to a meaningful outcome and allow testing the stated hypotheses?

Reviewer #1: Partly

3. Is the methodology feasible and described in sufficient detail to allow the work to be replicable?

Reviewer #1: No

4. Have the authors described where all data underlying the findings will be made available when the study is complete?

Reviewer #1: Yes

5. Is the manuscript presented in an intelligible fashion and written in standard English?

Reviewer #1: Yes

6. Review Comments to the Author

You may also provide optional suggestions and comments to authors that they might find helpful in planning their study.

Reviewer #1: This article represents a protocol for a systematic review on an important topic in developing countries. The authors utilized the PRISMA statement to establish a standard protocol for their study.

-Line 48: It is more appropriate to say: mother to fetus instead of “mother to child”

-Lines 55-57: “In Ethiopia, … respectively”. This sentence seems a bit unclear. Please rewrite it more transparently. A native English language editor can resolve minor issues in the use of the English language in this manuscript.

-Lines 77-78, 80, 124, 125, 209, 213, 215: please use the acronym instead of “sexually transmitted infections”.

-Line 129: “CINAH”, did you mean CINAHL?

-Line 129: Please indicate how you will screen the results from google and google scholar. For example, indicate if you want to screen a limited number of studies in these search engines. Usually, searching in google and google scholar leads to numerous studies, which screening all of them is not feasible.

-Lines 128-145 and additional file 2: Some alterations can help to improve the quality and feasibility of the search strategy:

I think the presented search strategy is too broad and will result in numerous irrelevant studies. For example, based on additional file 2 and using Term 5 to search in PubMed, there will be more than 812 thousand results, which is definitely too high to be screened. I do recommend consulting with an expert to develop a more specific search strategy.

Also, when you use “prevention & control OR control” it seems redundant. Because when you search “control” it automatically contains “prevention & control”. Another similar example is “Condoms OR Condoms use”. Please revise your search strategy in this regard.

You used “Unsafe Sex”, two times in Term #1.

In addition, I suggest mentioning the search strategy only in additional file 2, and removing it from the main text to prevent redundancy.

7. PLOS authors have the option to publish the peer review history of their article (what does this mean?). If published, this will include your full peer review and any attached files.

Reviewer #1: No

---

## [Author Response · Author response to Decision Letter 0]

7 Dec 2021

Response to reviewer 

#1

Mother to fetus (line 48)

Women and men … respectively (line 56-57.)

STIs (line 76-77, 124, 126, 14-143, 182, 191, 195, 199, 201 )

CINAHL (Line 129 )

As searching in Google and Google scholar leads to numerous studies, limited number of studies will be screened from Google and Google scholar using this phrase “Preventive practices towards Sexually Transmitted Infection in Ethiopia” (line 130-132)

Search strategy (additional file 2) (line 135)

Filter will be applied on PubMed database on titles/abstract, full text, journal articles, observational studies, and English language to manage the available studies (additional file 2).

---

## [Decision Letter · Decision Letter 1]

28 Dec 2021

PONE-D-21-25527R1Preventive practices toward Sexually Transmitted Infections and its determinants among young people in Ethiopia: a protocol for systematic review and Meta-analysisPLOS ONE

Dear Dr. Anbesu,

Thank you for submitting your manuscript to PLOS ONE. After careful consideration, we feel that it has merit but does not fully meet PLOS ONE’s publication criteria as it currently stands. Therefore, we invite you to submit a revised version of the manuscript that addresses the points raised during the review process.

We look forward to receiving your revised manuscript.

Kind regards,

Ali Rostami

Academic Editor

PLOS ONE

Journal Requirements:

Reviewers' comments:

Reviewer's Responses to Questions

**Comments to the Author**

1. Does the manuscript provide a valid rationale for the proposed study, with clearly identified and justified research questions?

Reviewer #1: Yes

2. Is the protocol technically sound and planned in a manner that will lead to a meaningful outcome and allow testing the stated hypotheses?

Reviewer #1: Yes

3. Is the methodology feasible and described in sufficient detail to allow the work to be replicable?

Reviewer #1: Yes

4. Have the authors described where all data underlying the findings will be made available when the study is complete?

Reviewer #1: Yes

5. Is the manuscript presented in an intelligible fashion and written in standard English?

Reviewer #1: No

6. Review Comments to the Author

You may also provide optional suggestions and comments to authors that they might find helpful in planning their study.

Reviewer #1: The authors have addressed most of my comments. The protocol is now technically sound, however, the manuscript still needs a revision in terms of English language use, and some alterations are necessary:

The authors want to search four databases: PubMed, CINAHL, Google, and Google Scholar. I recommend adding "Scopus" to widen the scope of your literature search.

Line 20: This sentence is unclear “: Globally, about 178.5 million new cases of curable sexually transmitted infections occurred among young people aged 15-24 years.” OR in line 52, 53: “Globally, about 357 million new cases of curable sexually transmitted infections occurred among people aged 15–49 years, and over half of them were between 15-24 years ”

Are these sentences about the estimated "annual" new cases? If so, the authors should mention.

Line 26: Change CINAH to “CINAHL” in the abstract.

Line 43, 54, 63, 74: Abbreviations should be defined in the text just at first use. You repeated Sexually transmitted infections in the mentioned lines.

Line 57: Change “STIs causes” to “STIs cause”

Line 46: Abbreviations should be defined in the text at first use: human immunodeficiency virus (HIV)

Line 65: Do you mean “contraceptive methods”?

Line 73: STIs “remain”

Line 145, 146: “However, for studies that examined both quantitative and qualitative study” change to “both quantitative and qualitative data”

Line 148: Would you please explain why you want to exclude “national surveys”? For example, why do you want to exclude an Ethiopian national survey on this topic?

7. PLOS authors have the option to publish the peer review history of their article (what does this mean?). If published, this will include your full peer review and any attached files.

Reviewer #1: No

---

## [Author Response · Author response to Decision Letter 1]

3 Jan 2022

Response to reviewer 

#1

Scopus data base (line 28, 127 )

Globally, the estimated annual new cases of curable sexually transmitted infections occurred among young people aged 15-24 years was about 178.5 million ((line 20-21 ). 178.5 million new cases was just to estimate about half of the 357 million cases among people aged 15–49 years were between 15-24 years. i.e 50% of 357 ≈178.5 (line 55)

Globally, the estimated annual new cases of curable sexually transmitted infections occurred among people aged 15–49 years was 357 million, and about half of them were between 15-24 years (line 54-55). 

CINAHL (line 27)

STIs (line 45, 54, 56, 64, 74, 86,88, )

human immunodeficiency virus (HIV) (line 48)

STIs cause (line 59)

contraceptive methods (line 67)

STIs remain (line 73)

Both quantitative and qualitative data (line 143)

National survey. It was mistake and corrected now (line 145)

---

## [Editor Report · Decision Letter 2]

11 Jan 2022

Preventive practices toward Sexually Transmitted Infections and its determinants among young people in Ethiopia: a protocol for systematic review and Meta-analysis

PONE-D-21-25527R2

Dear Dr. Anbesu,

We’re pleased to inform you that your manuscript has been judged scientifically suitable for publication and will be formally accepted for publication once it meets all outstanding technical requirements.

Kind regards,

Ali Rostami

Academic Editor

PLOS ONE

Additional Editor Comments (optional):

Thank you for revisions. The manuscript is acceptable. There is only one issue. Please use full term for STIs in beginning of introduction.

Best regards
---

## [Editor Report · Acceptance letter]

26 Jan 2022

PONE-D-21-25527R2 

Preventive practices toward sexually transmitted infections and their determinants among young people in Ethiopia: a protocol for systematic review and meta-analysis 

Dear Dr. Anbesu:

I'm pleased to inform you that your manuscript has been deemed suitable for publication in PLOS ONE. Congratulations! Your manuscript is now with our production department. 

Kind regards, 

on behalf of

Dr. Ali Rostami 

Academic Editor

PLOS ONE